
# The Kerala flood of 2018: combined impact of extreme rainfall and reservoir storage

Vimal Mishra[1*], Saran Aadhar[1], Harsh Shah[1], Rahul Kumar[1], Dushmanta Ranjan Pattanaik[2], Amar Deep Tiwari[1]

[1]Civil Engineering, Indian Institute of Technology Gandhinagar, Gandhinagar, 382355, India
[2]India Meteorological Department, New Delhi, 110003, India

*Correspondence to*: Vimal Mishra (vmishra@iitgn.ac.in)

**Abstract.** Extreme precipitation events and flooding that cause losses to human lives and infrastructure have increased under the warming climate. In August 2018, the state of Kerala (India) witnessed large-scale flooding, which affected millions of people and caused 400 or more deaths. Here, we examine the return period of extreme rainfall and the potential role of reservoirs in the recent flooding in Kerala. We show that Kerala experienced 53% above normal rainfall during the monsoon season (till August 21st) of 2018. Moreover, 1, 2, and 3-day extreme rainfall in Kerala during August 2018 had return periods of 75, 200, and 100 years. Six out of seven major reservoirs were at more than 90% of their full capacity on August 8, 2018, before extreme rainfall in Kerala. Extreme rainfall at 1-15 days duration in August 2018 in the catchments upstream of the three major reservoirs (Idukki, Kakki, and Periyar) had the return period of more than 500 years. Extreme rainfall and almost full reservoirs resulted in a significant release of water in a short span of time. Therefore, above normal seasonal rainfall (before August 8, 2018), high reservoir storage, and unprecedented extreme rainfall in the catchments where reservoirs are located worsened the flooding in Kerala. Reservoir operations need to be improved using a skilful forecast of extreme rainfall at the longer lead time (4-7 days).

## 1 Introduction

Extreme precipitation events, landslides, and floods are the most common natural disasters that affect human society and economy (Coumou and Rahmstorf, 2012; Crozier, 2010; Hirabayashi et al., 2008; Roxy et al., 2017). Frequent extreme precipitation events cause flooding (Fowler et al., 2010), which have become common in India (Mohapatra and Singh, 2003). The frequency of great floods and extreme precipitation events has substantially increased under the warming climate, which is consistent with the observations as well as climate model projections (Ali and Mishra, 2018; Milly et al., 2002). India has witnessed some of the most unprecedented extreme precipitation events that caused flooding and loss of lives in the recent past. For instance, extreme precipitation in Uttarakhand in 2013 resulted in large flooding with the death of more than 6000 people and economic loss of more than 3.8 billion USD ("Rapidly Assessing Flood Damage in Uttarakhand, India," 2014). Houze et al. (2017) argued that the 2013 extreme precipitation event was different than the historic flood producing events that occurred in the Himalayan region and was not caused by the convective storm. The heavy rain event



occurred in 2015 caused flooding in Chennai and led to the estimated damage of $3 billion (van Oldenborgh et al., 2016). Similarly, heavy rain in Mumbai in 2005 caused the death of more than 1000 people (Kumar et al., 2008). Extreme precipitation and flooding have become among the costliest natural disasters in India and other regions of the globe (The Human Cost of Natural Disasters 2015: A Global Perspective). Human losses from flooding are projected to increase by 70-

5 80% if the global mean temperature increases above 1.5°C from the pre-industrial level (Dottori et al., 2018). Moreover, Dottori et al., (2018) reported that the future flood impacts are likely to have uneven regional distribution, with the highest losses are to occur in Asia.

The recent extreme rainfall and widespread flooding in Kerala exemplify the enormity of extreme rainfall and large-scale floods in India. The persistent and extreme rainfall occurred in August 2018 in Kerala affected all the aspects of human lives

10 including socioeconomic conditions, transportation, infrastructure, agriculture, and livelihood. The Kerala flood of 2018 has already attracted attention from the media, scientific community, and policymakers, which is probably the worst flood in a century (The Independent, 16 August 2018). As per the preliminary estimates, the Kerala flood caused the death of more than 440 people (Gulf News, 30th August 2018) and economic damage exceeding $3 billion (News18, 17 August 2018). Despite the state-wide extreme rainfall in Kerala in August 2018, potential causes (heavy rain and reservoir operations) of

15 floods have been greatly debated. Here, we provide the first assessment of the anomalous nature of extreme rainfall and the role of major reservoirs that might have contributed in the flooding.

## 2 Data and Methods

We obtained daily observed rainfall data from India Meteorological Department (IMD) at 0.25° for the period 1901-2018 (till 21st August). IMD rainfall dataset is developed using more than 6000 observing gauge stations across India, and a

20 substantial number of stations are located in Kerala. Station based rainfall observations were interpolated using Inverse Distance Weighting interpolation (Pai et al., 2014; Shepard, 1968). The gridded dataset captures climatological, orographic, and other features associated with the Indian summer monsoon rainfall over India (Mishra et al., 2014; Shah and Mishra, 2015). Moreover, heavy rainfall variability over the Western Ghats and foothill of Himalaya region is well represented (Pai et al., 2014). Gridded daily rainfall from IMD has been widely used in hydro-climatic studies as well as for analysis of

25 extreme rainfall over India (Ali et al., 2014; Ali et al., 2018; Kumar et al., 2013; Mishra et al., 2016; Shah & Mishra, 2015). Using the daily rainfall gridded data, we estimated mass curve (the relationship between cumulative rainfall and time) and depth-duration- frequency (DDF) curve to evaluate the severity and intensity of extreme rainfall during August 2018 in Kerala. DDF curves are essentially similar to the intensity-duration-frequency (IDF) curves but with rainfall depth instead of intensity. The mass curve and DDF curves are widely used to extract the information of a storm event (Borga et al., 2005;

Overeem et al., 2009).

One of the important questions that the Kerala (Fig. S1) flood event poses is if the extreme rain event was once in a century (with the probability of exceedance 1% or less) event or not. Therefore, we estimated return period of heavy rain occurred in





August 2018 for 1-15 days durations. We used Generalized Extreme Value (GEV) distribution (Katz et al., 2002, 2005), which has been widely applied to analyze extreme hydro-climatic events (Ali et al., 2014; Min et al., 2011; Mishra et al., 2016). We fit the GEV distribution to annual maximum rainfall for the selected durations. The GEV distribution has three parameters (location, scale, and shape) that were estimated using the Maximum Likelihood method. Using the GEV

distribution, we developed DDF curves for 1-15 days extreme rainfall duration for 20, 50, 100, 200, and 500 years return period. We tested the goodness of fit of GEV distribution using QQ plots (Fig. S2) and Chi-Square test, and we found that the GEV distribution passed the goodness-of-fit test for the most of the rainfall durations (Table S1). Furthermore, we also estimated the anomaly of cumulative rainfall during the flood to quantify the rainfall departure from its mean estimated for the reference period of 1901-2017.

To analyze the reservoir condition before the flood event, we obtained reservoirs storage information for the seven major reservoirs (Idukki, Idamalyar, Kakki, Kallada, Malampuzha, Parambikulam, and Periyar) from India Water Resources Information System (WRIS) (IWRIS, www.india-wris.nrsc.gov.in) for the period 2007-2018. Using the IWRIS data, we analyzed the abnormality in the reservoir storage before the flood in Kerala. There is a number of small and medium size reservoirs in Kerala; however, as their storage information is not available, we mainly focus on the major reservoirs.

## 3 Results

First, we analyze the mass curve of rainfall between May 1 and April 30 for the period of 1901-2017 (Fig. 1a). We considered the period from May 1 as the onset of the summer monsoon happens mostly in May in Kerala. We find that the long-term (1901-2017) average annual rainfall in Kerala is about 2400 mm with a standard deviation of 400 mm. The 117 years observed record of rainfall in Kerala shows that the two wettest years occurred in 1924 and 1961 with the annual

rainfall of about 3600 mm (Fig. 1a). Rainfall during August 2018 in Kerala departs significantly from the averaged mass-curve. However, the 2018 rainfall (till August) in Kerala is lower than the observed rainfall in 1924 and 1961.

We estimated DDF curves for extreme rainfall averaged over the entire state of Kerala for the 1-15 days duration and 50-500 year return period using GEV distribution fitted on annual maximum rainfall for the 1901-2017 period (Fig 1b, Table S2, and S3). The 117 years record of observed rainfall provides us a basis to estimate the return period of extreme rainfall during

August 2018 (Table S3 and S4). We find that the return period for the extreme rainfall in 2018 varies with the duration (Fig. 1b, Table S3). For instance, 1-day maximum rainfall (120.2 mm, Table S4) in August 2018 (occurred on 15 August) in Kerala had a return period of about 75 years (Fig. 1b, Table S3). Moreover, the 2-day maximum rainfall (235.5 mm, Table S4) in August 2018, which occurred on 14-15 August, had a return period of about 200 years (Table S3 and Fig 1b). The 3-day maximum rainfall (294.2 mm, Table S4) in August 2018 was more than a 100-year event considering the record of 117

30    years. The return period of 10-day maximum rainfall (591.8 mm) in August 2018 was about 50 years (Table S3, S4 and Fig. 1b). Our results show that the 2-day maximum rainfall in Kerala during August 2018 was the most detrimental to the return period of more than 200 years.




The monsoon season of 2018 has been anomalously wet as the majority of the state received more than 1500-2000 mm rainfall during May 1 to August 21 (Fig. S3). Rainfall in 2018 exceeded by more than 40% in most of Kerala while the southern region experienced more than 200% rainfall between May 1 and August 21, 2018 (Fig. S3). We also analyzed the cumulative rainfall in Kerala during August 8 to August 17, 2018, when extreme rainfall occurred in large part of the state (Fig. 1c). We find that during these 10 days a majority of the state received more than 500 mm rainfall with a surplus of 40-50% (Fig. 1d). Rainfall during the 10-day period (August 8-August 17) departed more than three standard deviation away from mean (Fig. 1d). Analysis of daily rainfall from August 3-20 shows that substantial rainfall occurred during 8-9 August 2018 (Fig. S4, S5, and S6), which continued till August 18 in Kerala (Fig. S4 and S5). Rainfall occurred on August 15, and 16 was anomalously higher as a large part of Kerala received more than 200 mm rainfall each day (Fig. S4). As shown by our results that persistent rainfall before 15-16th August might have created saturated conditions before heavy rain that caused enormous flooding and loss of lives in Kerala.

To further diagnose the causes of flooding in Kerala, we analyzed the storage of major reservoirs before the flooding (on 8th August 2018). We examined the storage condition of the seven (Idukki, Idamalyar, Kakki, Kallada, Malampuzha, Parambikulam, and Periyar, Fig. S1) major reservoirs. We find that reservoir storage before the flooding was anomalously high in most of these seven reservoirs (Fig. 2). All the seven major reservoirs had storage much higher than their long-term (2007-2017) mean (Fig. 2). For instance, six (out of seven) reservoirs were at more than 90% of their full reservoir level (FRL) on August 8, 2018 (Fig. 2, Table S5). Parambikulam had reservoir storage 99.5% of its FRL while in Idukki, Idamalyar, Kakki, Kallada, Malampuzha reservoir storage on August 8, 2018, was 92.5, 97.3, 90.5, and 97.8% of their FRL (Table S5, Fig. 2). The only major reservoir that had reservoir storage of less than 80% of its FRL was Periyar (Table S5). Surplus (40-50%) rainfall occurred during May 1 to August 2018 in Kerala has contributed to above normal reservoir storage. Kerala has many other small and medium size reservoirs that might have had high storage before the flooding. However, we consider only the major reservoirs that have long-term storage data for our analysis. Our results based on reservoir storage show that most of the major reservoirs were almost full before the extreme rainfall occurred on 14-16 August 2018. Therefore, reservoirs did not have the capacity to accommodate the additional flow generated by extreme rainfall. Due to extreme rainfall after August 8, reservoirs had to release a substantial amount of water in a short span of time. A red alert was flashed on 9th August 2018 on Idukki reservoir, which later on opened all the gates to release water to lower the reservoir level (India Today, August 9, 2018). The combination of heavy rain and reservoir release might have worsened the flood condition in Kerala.

Next, we analyzed rainfall occurred in the catchments upstream of the major reservoirs (Table S6 and S8). Here our interest is to diagnose if the extreme rainfall occurred upstream of these major reservoirs was also unprecedented (Fig. S8, S9, and S10). To do so, we constructed mass curves using cumulative rainfall for 1901-2017 for May 1 to April 30(Fig. 3). We compared the mass curve of 2018 (for May 1 to April 30) against the historic data. Our analysis based on the mass curve for



the catchments upstream of the reservoirs shows that 1924 and 1961 were the two most anomalous years in the entire record of 117 years (Fig. 3). However, rainfall occurred in 2018 (May 1 to August 21) was unprecedented in the last 118 years for Idukki, Kakki, and Periyar reservoirs (Fig. 3, Table S8). Idukki, Kakki, and Periyar reservoirs received 279, 700, and 420% higher rainfall from their long-term mean (for the same period) in 2018 (Table S6). Furthermore, Kallada reservoir received

195% higher rainfall from its long-term (1901-2017) mean (Table S6, Fig. 3). The extremely wet rainfall season during the monsoon of 2018 (before August 8) might have resulted in the above normal flow to the reservoirs, which in turn made the majority of the major reservoirs almost full (higher than 90% of their FRL) before the extreme rainfall occurred during 14-16 August in Kerala. Our results highlight that state-averaged rainfall in 2018 was only 53% higher than its long-term mean. However, rainfall occurred in the catchments upstream of the major reservoirs was significantly higher than just 53%. Above

normal seasonal rainfall, high reservoir storage, and unprecedented rainfall in the catchments where reservoirs are located created a most favorable situation for massive flooding in Kerala.

Finally, we constructed DDF curves to understand anomalies of extreme rainfall (1-15 days duration) occurred in the catchments upstream of the major reservoirs (Fig. 4). We notice a substantial spatial variability in extreme rainfall in the

catchments upstream of the major reservoirs. For instance, 1-day maximum rainfall for Idukki catchment has a return period of about 300 years (Fig. 4a and S8). However, extreme rainfall for 2-15 days duration in the Idukki catchment has a return period significantly higher than 500 years (Fig 4a, Table S7, and S8). Similarly, Kakki and Periyar catchments experienced extreme rainfall that had more than 500 years return period. Moreover, extreme rainfall for 10 or more days in the catchment upstream of Kallada reservoir had a return period higher than 500 years (Fig 4d, Table S7, and S8). Therefore, the

catchments upstream of the major reservoirs experienced unprecedented extreme rainfall in August 2018 in the entire record of 117 years. Moreover, extreme rainfall in these catchments had return periods much higher than state averaged heavy rain in Kerala. As most of these reservoirs were at their maximum storage capacity before the extreme rainfall occurred, the combination of reservoir storage and persistent extreme rainfall together caused the unprecedented flooding during August in Kerala.

**4 Discussion**

The frequency and intensity of extreme precipitation events have increased in India during the last few decades (Goswami et al., 2006; Guhathakurta et al., 2011; Rajeevan et al., 2008; Roxy et al., 2017). However, if the Kerala flood event can be attributed to climate change or not remains the research question for future. Recently, Mukherjee et al. (2018) used data from the coupled model intercomparison project 5 (CMIP5) and C20C+ detection and attribution project and reported an increase

in extreme precipitation under anthropogenic warming in India. They (Mukherjee et al., 2018) found that 1-5day extreme precipitation at 5-500 year return period increases by 10-30% under the anthropogenic warming. Moreover, extreme precipitation in southern India increases with a much faster (18%/K) rate in response to warming in comparison to north



India (Mukherjee et al., 2018). However, the attribution of a single extreme event to climate change is difficult, despite the consensus of the increase in extreme precipitation under the warming climate. For instance, van Oldenborgh et al. (2016) reported that climate change did not cause the Chennai flooding and heavy rain event in 2015.

Our results highlight that the Kerala flood was caused by multi-day extreme rainfall and partly due to high reservoir storage. For instance, the state averaged 2, and 3-day extreme rainfall had return periods of more than 200 and 100 years, respectively. However, persistent and heavy rain in Kerala during 8-17 August may not be the only reason for the large-scale flooding. The role of other factors such as land use land cover change (Blöschl et al., 2007; Wheater and Evans, 2009) in the Kerala flooding remains to be examined. However, we find that the reservoir storage might have played a major role in

worsening the flood situation in the state. Based on the observed record, we find that almost all the major reservoirs were more than 90% full before the heavy rain (14-17 August 2018). Moreover, heavy rain in the catchments upstream of the major reservoirs was associated with a much larger return period that the state had witnessed. For instance, heavy rain for four out of seven reservoirs had more than 500 years return period. Apart from the role of heavy rain and reservoirs, Kerala received above normal rainfall during the monsoon season of 2018, which might have also contributed to flooding.

The 2018 event in Kerala shows that reservoirs can play a major role in improving or worsening the flood situation. Reservoir storage during the monsoon season plays a major role in providing water for irrigation and hydropower generation. Therefore, most of the reservoirs maximize their storage before the monsoon departs, as we have observed for the major reservoirs in Kerala. Reservoir operations can be more effective by incorporating the extended range extreme rainfall

forecasts (Hossain et al., 2010; Pattanaik and Das, 2015).

## 5 Conclusions

Based on our findings, the following conclusions can be made:

1.  We analyzed mean and extreme rainfall in Kerala during 1901-2018. We find that Kerala received more rainfall in 1924 and 1961 than 2018 between May 1 and August 21. Kerala received 53% more rainfall between May 1 and
August 21 in 2018 from its long-term mean for the same period.

2.  We estimated return period of extreme rainfall in Kerala during August 2018 using GEV distribution. We find that 1-day maximum rainfall averaged over the entire state in August 2018 had a return period of about 75 years. However, 2 and 3-day maximum rainfall had the return period of about 200 and 100 years, respectively.

3.  Most of the major reservoirs in Kerala had more than 90% of its capacity on 8th August 2018. Since heavy rain in
the catchments upstream of major reservoirs was unprecedented, the reservoirs had to release the considerable amount of water in a short span of time. For instance, Idukki, Kakki, and Periyar reservoirs, which were already almost full, witnessed extreme rainfall of more than 500 years return period. Idukki, Kakki, and Periyar reservoirs





received 279, 700, and 420% surplus rainfall, respectively from their long-term mean between May 1 and August 21 in 2018.

4. Kerala received above normal rainfall in the monsoon season of 2018, which contributed to reservoir storage significantly. The combination of above normal seasonal rainfall, state-wide extreme rain, high reservoir storage, and unprecedented extreme rain in the catchments upstream to major reservoirs might have played a significant role in the large-scale flooding in Kerala. Seasonal and extended range forecast of rainfall and improved forecast of extreme rain events at a longer lead can help in reservoir operations.

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





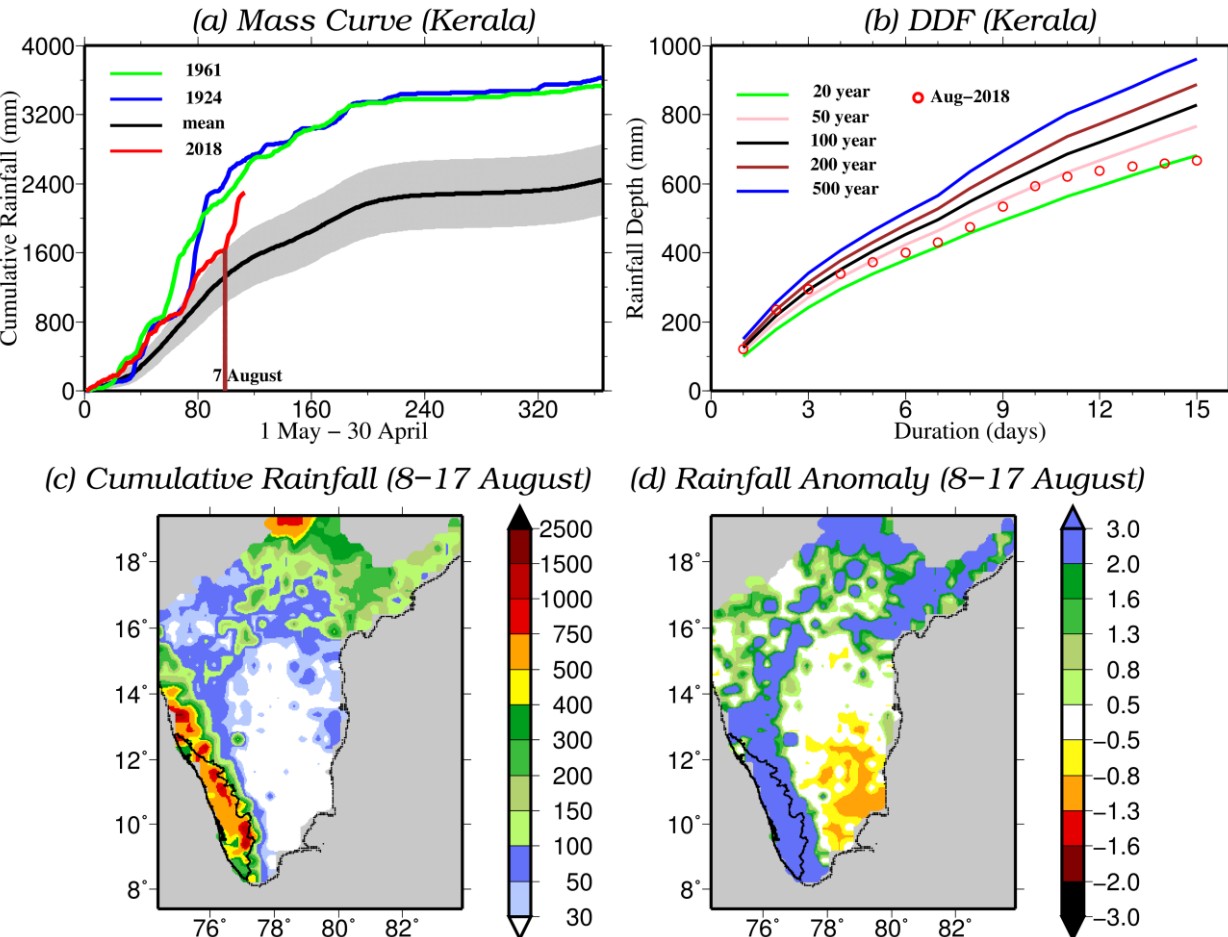

**Figure 1: (a) Mean mass curve (black line) of daily rainfall for May 1 to 30 April for 1901-2017 period. The shaded area shows variability estimated using one standard deviation. Red, blue, and green lines show extreme rainfall during 2018, 1961, and 1924, respectively. (b) Depth-duration-frequency curves for extreme rainfall in Kerala for 20-500 year return period estimated using Generalized Extreme Value (GEV) Distribution fitted for the 1901-2017 period. Red circles show extreme rainfall for 1-15 days during August 2018 (c) Cumulative rainfall in Kerala during 8-17 August 2018, and (d) rainfall anomaly (standard deviation) for 8-17 August estimated against cumulative rainfall for the same period during 1901-2017.**





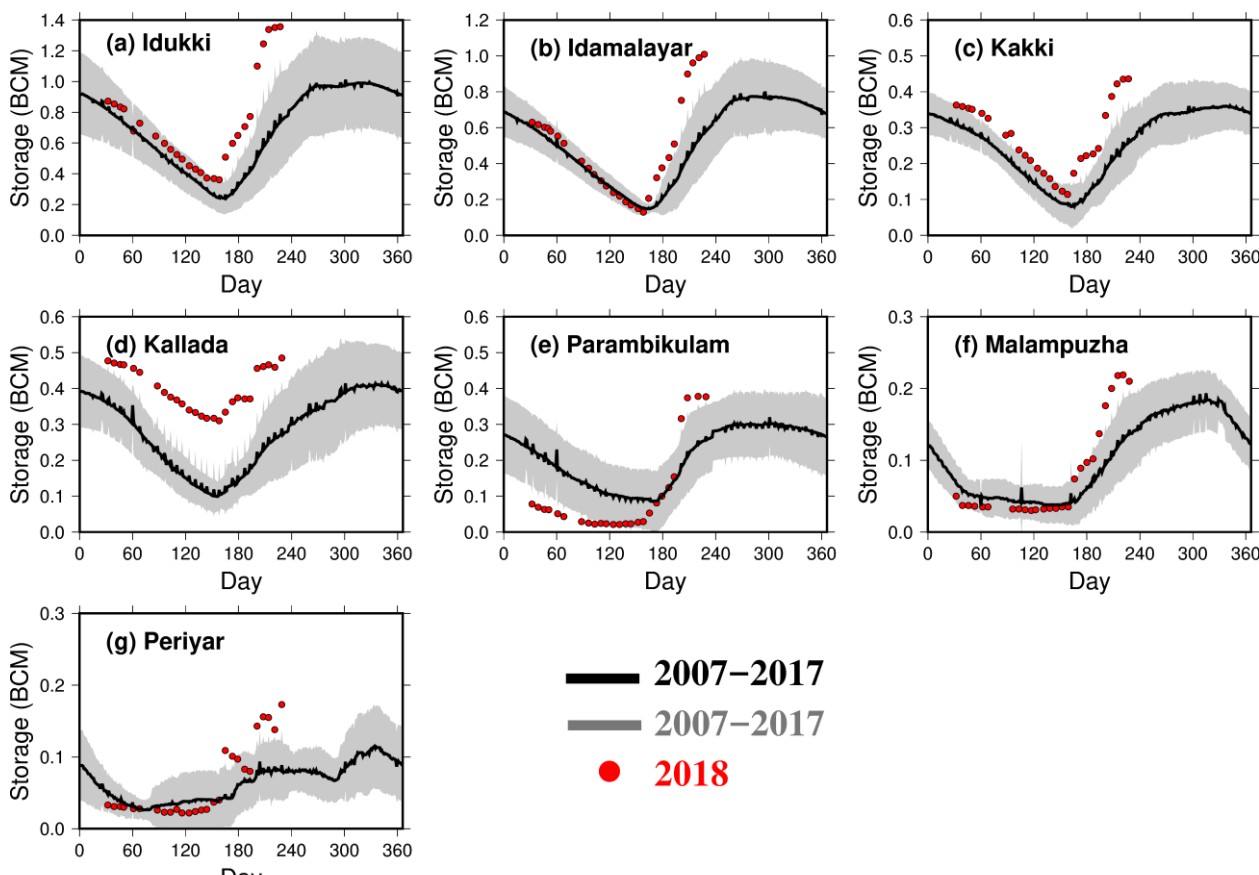

**Figure 2: Reservoir storage (billion cubic meters: BCM) during August 2018. (a) Daily mean reservoir storage and variability estimated using one standard deviation (shaded part) for the 2007-2017 period for Idukki reservoir, (b-g) same as (a) but for Idamalayar, Kakki, Kallada, Parambikulam, Malampuzha, and Periyar reservoirs. Reservoir storage for 2018 is available from January 1 to August 8, 2018, from Water Resources Information System (WRIS).**





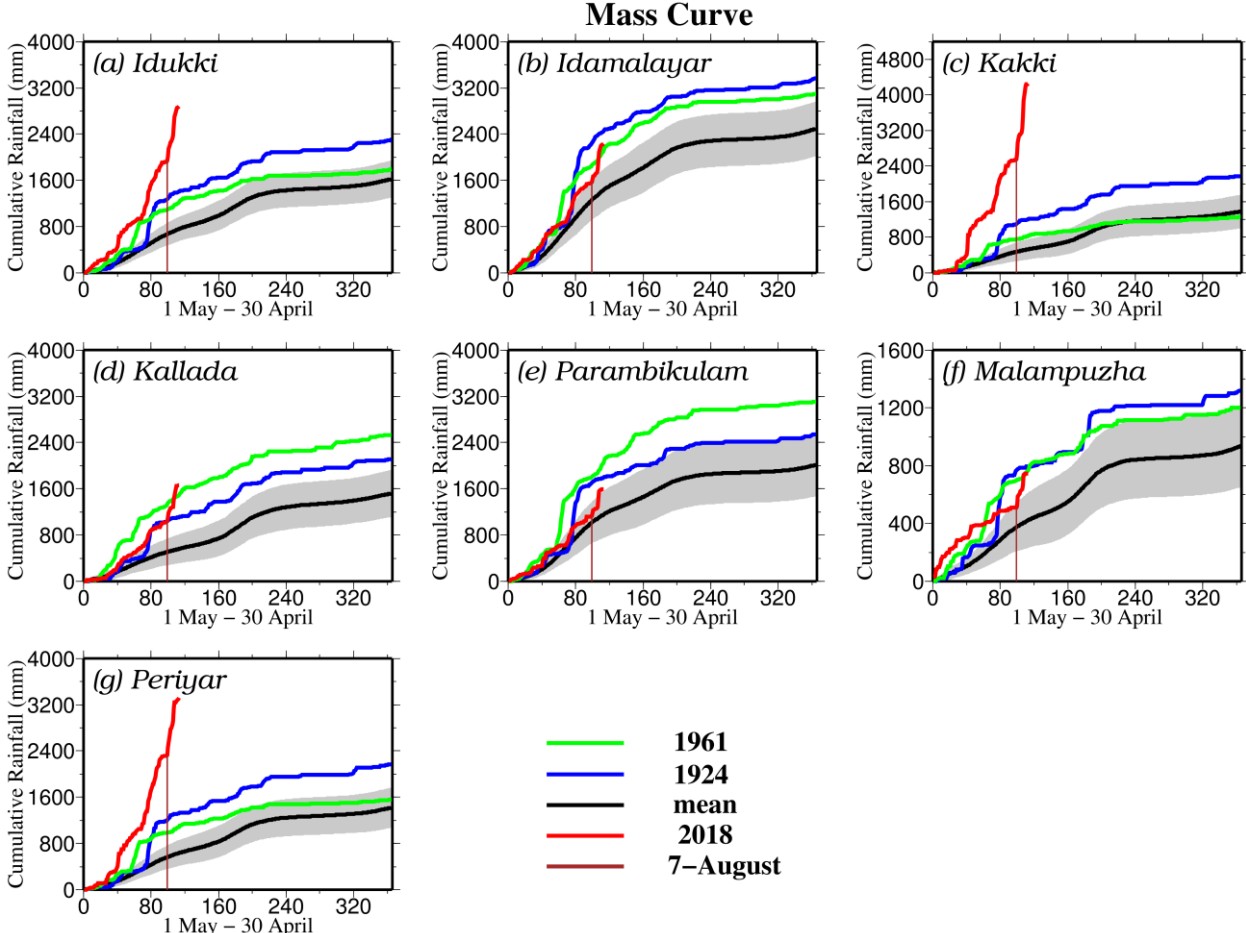

**Figure 3: Mass curves estimated using cumulative rainfall aggregated for the catchments upstream of the major reservoirs located in Kerala. Black line shows mean mass curve for 1901-2017 period while the shaded region shows variability in mass curves estimated using one standard deviation for 1901-2017. Red, green, and blue lines represent mass curves for 2018, 1961, and 1924, respectively. The vertical line shows mass-curve on August 7 2018.**





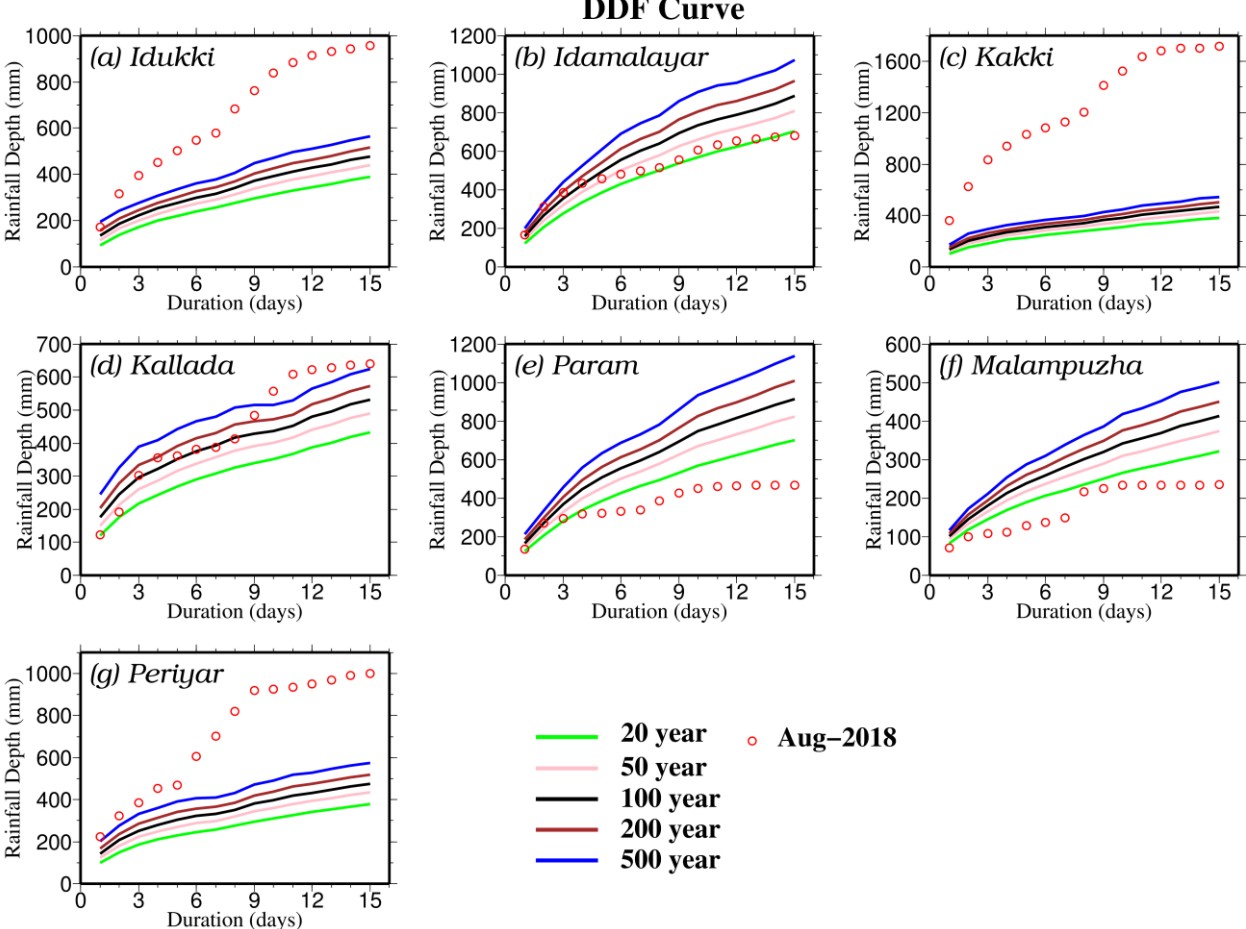

**Figure 4: Depth-duration-frequency curves for extreme rainfall in all 7 reservoir catchments for 20-500 year return period estimated using Generalized Extreme Value (GEV) Distribution fitted for the 1901-2017 period. Red circles show extreme rainfall for 1-15 days during August 2018.**

