# Peer review of "The Kerala flood of 2018: combined impact of extreme rainfall and reservoir storage"

_Hydrology and Earth System Sciences, 2018_

## Referee Comment (RC1) · S. Padikkal (Referee) · 27 Sep 2018

This paper and its findings are very relevant in the context of extreme climatic events recurring at an alarming rate all over the world. A critical analysis of Kerala flood of 2018 presented here must be of great interest to the researchers, water resources managers as well as the development practitioners. Authors have done a good job and the presentation of its results are appealing to the scientific community.

The estimation of return period of extreme rainfall using GEV distribution and its analysis are well presented. In that aspect, this paper is an outstanding work. However, there are certain important aspects missed out in the analysis of reservoirs' storage and that makes the conclusions drawn a bit unscientific. I would request the authors to

consider the following aspects and rework on it to make the paper more applicable to a wider section of readers among researchers and practitioners.

(a) Unlike the impact of extreme rainfall analysed and presented, the impact of reservoir storage in worsening the flood situation appears to be only a general statement and it is not supported with a clear and logical analysis. (b) Assessing the impact of reservoir storage by just looking at the percentage of FRL storage is a premature analysis.

This has to be analysed in a more logical manner by checking the rise in water level caused at a downstream point due to reservoir releases. To be very specific, you need to do this analysis for at least 5 major reservoirs you are considering. Idukki, Idamalayar and Periyar in the Periyar basin, Malampuzha in the Bharathapuzha basin and Parambikulam in the Chalakkudipuzha basin. CWC gauging stations are available on the downstream side of these reservoirs in Periyar, Chalakkudipuzha and Bharathapuzha. So the flow data at these points are already there in the public domain.

Simulate the depth of flow at these points without the reservoir releases and with reservoir releases and see whether the impact is significant (reservoir release data is also in the public domain, or you can get it from the respective dam managers). If you are getting the impact significant, simulate again the reservoir storage to safer position and see the volume that should have been left unfilled in the reservoir to make the impact of its releases insignificant. This finding is also important. I would expect such an analysis before jumping into a conclusion that reservoirs storage also contributed to the flood.

Another important point to be considered in this analysis is that the main objective of major reservoirs you have taken is water conservation and not flood control. When you optimize the storage based on the major objective of water conservation, what is the limitation on unfilled volume should also be clearly investigated before making a conclusion.

Hope the authors would consider these points and revise the MS with these results to

make it more convincing. This paper would be of interest to the broader public if the authors take up the additional analysis I have suggested and revise the MS accordingly.

---

## Short Comment (SC1) · 26 Oct 2018

Note to the editor and authors: As part of an introductory course to the Master programme Earth Environment at Wageningen University, students get the assignment to review a scientific paper. Since several years, students have been reviewing papers that are in open online discussion for HESS or BGS, and they have been asked to submit their reports to the discussion in order to help the review process. While these reports are written in the form of official (invited) reviews, they were not requested for by the editor, and we leave it up to the editor and authors to use these reports to their advantage. While several students were often asked to review the same paper, this was not done with the aim to provide the authors with much extra work. We hope that these reports will positively contribute to the scientific discussion and to the quality

of papers published in HESS. This report/review was supervised by dr. Ryan Teuling (teacher within the ITEE course at Wageningen University and also associated editor with HESS).

This study is aimed at finding the causes of the Kerala flood in August 2018. A gridded rainfall map of India was constructed for the analyses and data of major reservoirs in Kerala were gathered. Based on the available data reservoir characteristics, mass curves and depth-duration-frequency curves were constructed. Results showed unprecedented amounts of rainfall in catchments upstream of major reservoirs with return periods much larger than 500 years. It was also shown that most major reservoirs were almost at their full reservoir level at the start of the series of extreme rainfall events. It is concluded that both the high reservoir storage and extreme amounts of rainfall played a significant role in the large-scale flooding in Kerala. It is advised to improve forecasts of extreme rain at longer lead times to better manage reservoir operations. Research in this field has a high societal relevance since a better understanding of the causes of major floods, like in Kerala, can save future lives. A similar study, Sayama et al. (2012), also stresses the importance of this kind of analyses to better understand and prevent future floods. The study is also relevant since similar events are predicted to occur more often in India under the effect of global warming. However, in my opinion the authors did not conduct a sufficiently in depth study of this interesting case. Results are so extreme that validity of the data has to be questioned. Moreover, a proper discussion about these extreme outcomes is missing. Furthermore, some figures are missing or current figures need to be altered to improve the clarity of the paper. In spite of its high relevance, I would recommend that major revisions are necessary before the work can be published in HESS.

**Major comments**

(1) The paper concludes that rainfall in the catchments upstream of major reservoirs was unprecedented. It is stated that Idukki, Kakki and Periyar reservoir experienced 279, 700, and 420% surplus rainfall, respectively from their long-term mean between May 1 and August 21 in 2018. These values are so extreme that I highly doubt the correctness of the data. Moreover, the discussion section does not mention anything about these seemingly unrealistic results.

These conclusions are based on figures 3 and 4. Idukki, Kakki and Periyar reservoirs clearly stand out from the rest. In figure 3 the upstream catchments of Idukki, Kakki and Periyar reservoirs show cumulative rainfall amounts after three months that are already 2-3 times higher than the long-term average amount of rainfall for an entire year in these catchments. Furthermore, the cumulative amount of rainfall in these three catchments is far away from the grey area indicating the area within one standard deviation from the long-term mean. In figure 4 the upstream catchments of Idukki, Kakki and Periyar reservoirs show rainfall amounts during August 2018 for a duration of 15 days that are 2-3 times as high as the amount that would occur once every 500 years. Rainfall amounts depicted in figures 3 and 4 are so unprecedented that it is highly unlikely that these amounts actually occurred.

Major alterations are necessary to sufficiently improve the discussion section and the results shown in figures 3 and 4. I would recommend to reconsider the data used to construct the figures. From the methods it is not clear to me whether a quality control has been conducted on the raw data. If not the case, I would strongly recommend to do one as this might eliminate outliers that cause the seemingly impossible results in figures 3 and 4. The quality control used in Yatagai et al. (2012) and Haylock et al. (2008) might be appropriate methods. Also, the discussion section should be expanded with a discussion about these extreme results.

(2) Major conclusions are solely based on data obtained from rain gauges in Kerala.

No information is given about the location of these gauges. Moreover, I am missing information about the size of the catchments upstream of the major reservoirs and the location of rain gauges in these catchments and their surrounding areas. Not including this information gives an incomplete picture of where the data comes from and what the data actually describes. Validation of the data could help in understanding how questionable results, like addressed in my previous point, can occur.

To improve the clarity of the paper I would first of all advise to include a map of all rain gauges in India, like is given in Pai et al. (2014a). In this map Kerala should be clearly delineated. By including this map readers of the paper will be able to see the density and distribution of rain gauges in India and Kerala. Furthermore, I think it is essential that separate maps of all reservoirs and their upstream catchments should be included. In this map the location of rain gauges in the catchment and its surroundings should be clearly indicated. Including these maps would contribute in understanding how the rainfall data of the catchments is obtained.

(3) In the paper variability is estimated using one standard deviation. One standard deviation indicates the variability in which 68% of the values will fall. This is meaningless since values outside this variability cannot be regarded as extreme.

Indicating variability with two standard deviations would be an improvement. 95% of all values will fall within a variability of two standard deviations, values outside this variability can now be regarded as extreme. This especially holds for figures 1a, 2 and 3. For example, extending the variability indicated by the grey area in figure 2 to two standard deviations would cause most values of 2018 to fall within this variability. Values that are outside this area can now be regarded as extreme, this would add to the strength and clarity of the figure.

**Specific comments**

(1) Be consistent with writing down dates throughout the document.

(2) Use "that" before "occurred" across the document like in p2 line 1, 9 and 32.

(3) The statement made in p3 line 9 seems very extreme and I think these amounts of rainfall are highly unlikely to have occurred in large parts of Kerala. Also, when looking at the DDF curves this will result in gigantic return periods. Based on what is this statement based? I would remove this statement.

(4) p2, line 1: missing "that" before "occurred"

(5) p2, line 6: "Dottori et al. (2018)" instead of "Dottori et al., (2018)"

(6) p2, line 6: "an" before "uneven"

(7) p2, line 9: missing "that" before "occurred"

(8) p2, line 19: missing "the" before "IMD"

(9) p2, line 19: Cite Pai et al. (2014a) here since the dataset has been developed based on the methods described in this paper.

(10) P2, line 22: The wrong citations are used here. Mishra et al. (2014) only cites a paper which makes this statement, Shah and Mishra (2015) does not make this statement at all. The only correct reference here should be Pai et al. (2014b).

(11) P2, line 24 and 25: It is stated that "Gridded daily rainfall from IMD has been widely used in hydro-climatic studies" with five references to support this claim. However, all references are papers published by one of the authors. Add references which are not of one of the authors or do not state that it is "widely used".

(12) p2, line 26 and 27: "curves" instead of "curve"

(13) p2, line 31: "a" before "once"

(14) p2, line 32: "The" before "return"

[Figure]

(15) p3, line 6: "periods" instead of "period"

(16) p3, line 6: "the" before "GEV"

(17) p3, line 6: "a" before "Chi-square"

(18) p3, line 6: replace "test, and we" by "test. We"

(19) p3, line 7: remove the third "the"

(20) p3, line 12: "IWRIS" instead of "WRIS"

(21) p4, line 2 and 3: I would not use percentages, they are quite meaningless here in the way it is written down. Better use exact numbers or reformulate.

(22) P4, line 7: "the" before "mean"

(23) P4, line 8: remove last ","

(24) The statement made in p4 line 9 seems very extreme and I think these amounts of rainfall are highly unlikely to have occurred in large parts of Kerala. Also, when looking at the DDF curves this will result in gigantic return periods. Based on what is this statement based? I would remove this statement.

(25) p4, line 16: the reservoir dataset from 2007-2017 is referred to as a long-term mean, I would not call 11 years of data a long-term dataset.

(26) p4, line 14: Remove double spacing before "We"

(27) p5, line 3: "respectively" before "279"

(28) p5, line 23: "2018" before "in"

(29) p5, line 28: "the" before "future"

(30) p5, line 30: "1-5 day" instead of "1-5day"

(31) p5, line 31: remove "the"

[Figure]

(32) p6, line 8: "and" before second "land"

(33) p6, line 12: "than" instead of "that"

(34) p6, line 13: "periods" instead of "period"

(35) p6, line 19: "of" before "extreme"

(36) p6, line 26: "periods" instead of "period"

(37) p6, line 28: remove "the"

(38) p6, line 28: "periods" instead of "period"

(39) p6, line 29: "were at" instead of "had"

(40) p7, line 7: "time" before "can"

(41) p7, line 7: "improving" before "reservoir"

(42) Figure 1a: I would indicate the start of events on the 7th of August in the same way as in figure 3 with a thin line. Indication now looks like a weird dip in the data.

(43) Figure 1 caption: state that the delineated part in India is Kerala.

**References**

Haylock, M. R., Hofstra, N., Tank, A. K., Klok, E. J., Jones, P. D., New, M. (2008). A European daily high-resolution gridded data set of surface temperature and precipitation for 1950–2006. Journal of Geophysical Research: Atmospheres, 113(D20).

Pai, D. S., Sridhar, L., Rajeevan, M., Sreejith, O. P., Satbhai, N. S., Mukhopadhyay, B. (2014a). Development of a new high spatial resolution (0.25× 0.25) long period

(1901–2010) daily gridded rainfall data set over India and its comparison with existing data sets over the region. Mausam, 65(1), 1-18.

Pai, D. S., Sridhar, L., Badwaik, M. R., Rajeevan, M. (2014b). Analysis of the daily rainfall events over India using a new long period (1901–2010) high resolution (0.25× 0.25) gridded rainfall data set. Climate dynamics, 45(3-4), 755-776.

Sayama, T., Ozawa, G., Kawakami, T., Nabesaka, S., Fukami, K. (2012). Rainfall–runoff–inundation analysis of the 2010 Pakistan flood in the Kabul River basin. Hydrological Sciences Journal, 57(2), 298-312.

Yatagai, A., Kamiguchi, K., Arakawa, O., Hamada, A., Yasutomi, N., Kitoh, A. (2012). APHRODITE: Constructing a long-term daily gridded precipitation dataset for Asia based on a dense network of rain gauges. Bulletin of the American Meteorological Society, 93(9), 1401-1415.

---

## Referee Comment (RC2) · Amit Kumar (Referee) · 27 Oct 2018

The manuscript 'The Kerala flood of 2018: combined impact of extreme rainfall and reservoir storage' by Mishra and others includes very interesting argument on recent flood in the Kerala. The topic is very interesting and of great interest to the scientific communities working in the field of climatology and hydrology as well. However, certain portion of the manuscript needs substantial reworking before it can be referred in HEES. The research purpose, significance and objectives are clearly stated but not well organized. More detailed and accurate descriptions are required. I suggest some parts of abstract would fit better into the introduction. Moreover, authors should begin the abstract directly with results/findings, and/or state that the area received several anomalous rainfall storms throughout the past and recently experienced 53% above

normal rainfall (Monsoon season 2018).

The rainfall in the Kerala is predominately controlled by the South-west and North-east monsoons. The area has witnessed heavy losses to life and property by floods in almost all rivers of Kerala due to several rainstorms in the past as well. Despite the fact such information is missing in the text and fails to convey a clear and convincing introduction. I suggest that keep one paragraph in introduction about the history of floods in Kerala and discuss the conditions about reservoir operations during such rainfall events. The contents of this paper are valuable, but the authors should pay close attention to the data sets and results. Authors have used rigorous statistical methods to compare peak monsoon rainfall patterns during two time periods. The team looked specifically at rainfall during the month of August, which is the peak of the monsoon. Further, explain the percent (%) of rainfall over Kerala during June, July and 1st to 19 th of August, above selected normal. Catchment area of each sub-basin is believed to be calculated and should be included in the text, and therefore comparison of rainfall depths observed in different sub-basins and rest of the Kerala during event will be computed.

Rainfall received during the summer monsoon season contributes about 70–90% of annual rainfall over India. The intensity and magnitude of these floods are the manifestation of year to year variability of monsoon over India. Also, it has been recognized that such variability of ISM has a good teleconnective relation with El Nino Southern Oscillation (ENSO), North Atlantic Oscillation and climate extreme indices available. Therefore, team should look at the effects of teleconnection patterns (TPs) on the extremes of precipitation over Kerala. Whether, the patterns of extreme wet and dry spells during the monsoon season have changed in recent decades (1901-2018). An understanding of the teleconnection patterns associated with these events could benefit many people and policy makers in the state. It is essential to re-evaluate the operating criteria, guidelines that govern the storage and release functions of a reservoir in such extreme conditions. You would need to have much more discussion highlighting how

your results are relevant for climate change adaptation and disaster risk reduction in the region. This has not really been demonstrated in discussion or conclusions.

Can you expand on this? "Reservoir operations need to be improved using a skillful forecast of extreme rainfall at the longer lead time (4-7 days)". Your result shows that, if the reservoir had been below FRL, the flooding conditions would have not changed much due to the severe storm continued for 3-4 days. It would have been necessary to release from the reservoirs after 1st day of the extreme rainfall. Therefore, improved forecast of extreme rainfall from onset of the monsoon along with reservoir conditions (Inflow and outflow) must be reviewed and design accordingly. The probable maximum flood (PMF) is frequently revised as the required inflow design flood (IDF) until all the necessary safety conditions will be satisfied.

Results and conclusions are almost similar. Findings you infer from your data should be worked out more thoroughly in discussion section. Try to be more stringent when presenting your data and avoid repletion of similar sentences. Pleas find more details in the attached file.

The manuscript needs to substantially improve in English. I think manuscript need thorough revision for it to get to the standard that it deserves. I strongly encourage authors to read these comments without any pre-conceived notion because I think this should be published once revised appropriately.

Please also note the supplement to this comment:
https://www.hydrol-earth-syst-sci-discuss.net/hess-2018-480/hess-2018-480-RC2-supplement.pdf

**Supplement:**

[revised manuscript text omitted]

---

## Short Comment (SC2) · 28 Oct 2018

In the submitted manuscript authors made an attempt to analyze the rainfall and reservoir level in the monsoon months of prior August 2019. Apparently, many of the findings that authors have claimed are already available in public domain perhaps not in the same way the team has presented. I have included only some of the links below for reference.

1.      https://www.firstpost.com/tech/news-analysis/what-caused-the-kerala-floods-4993041.html     2.      https://indianexpress.com/article/research/year-1099-keralas-great-flood-of-1924-too-affected-same-areas-5317677/
3.      https://www.bloombergquint.com/kerala-floods/kerala-flood-

of-2018-less-intense-than-deluge-of-1924-so-why-was-damage-as-
great 4. https://timesofindia.indiatimes.com/india/in-just-20-days-
kerala-gets-highest-aug-rains-in-87-yrs/articleshow/65480279.cms 5.
https://www.indiatoday.in/india/story/why-kerala-fears-repeat-of-1924-havoc-in-2018-
rainfall-1315884-2018-08-16 6. https://scroll.in/article/890593/monsoon-trends-for-
many-in-kerala-this-years-rains-recalls-the-great-flood-of-99

Apparently, some of the articles appeared in print and electronic media made a more comprehensive and holistic overview of Kerala flooding, I assume, keeping the data analysis in the background. Please excuse me if I am being blunt or it reads harsh, as a reader of Hydrology and Earth System Science I would always seek a bit more scientific content with cutting edge hydrological analysis from a HESS article. The submitted manuscript is undeniably a good piece of work when I see it as a term project where grad students are asked to perform a quick analysis of rainfall pattern and reservoir levels to reason with the flooding, however, for a scientific paper it requires some more meat.

At its current form the submitted manuscript can be accepted in conference proceedings but for being considered as a potential HESS journal article it requires major revision with some additional analysis. To avoid overlap in comments made in 'Referee Comment 2' and 'Short Comment 1', I am not including the common queries here and only including those parts which I find missing or less emphasized in their observations.

Since, I am not the assigned referee of the paper authors are free to discard my comments, however, I would like to state few of my concerns that I would like to addressed by authors to make the manuscript more elaborate, scientific, and a good hydrology paper rather than being a mere data analysis.

Major comments:

Comment 1: The flood extent and inundation through hydrodynamic modeling approaches in the downstream of various reservoirs taken in the study will provide more

insight into the problem. While doing so if authors can present a comparative analysis of flood extent between two scenarios as follows. Scenario 1: Flood extent with actual reservoir levels. Scenario 2: Flood extent if the reservoirs would have been regulated properly before the heavy rainfall hit the catchment i.e., the best recommended practice. Though it will take some good amount of effort, it will add enormous value to manuscript.

Comment 2: Did authors selected the specific IMD grid in which reservoir falls to analyze the rainfall or utilize the catchment average rainfall calculated for IMD gridded rainfall. Gridded data is prepared with the varying network of rain gages hence there could be inhomogeneity in the time series. Data constraint can be accepted while doing such analysis over India scale however, for small catchments only station data should be preferred. Since, this is a very localized study, it would be more appropriate if authors use station data instead of gridded rainfall data. Authors should include the rain gauge network from the reservoir catchment and utilize the rain gage data instead of gridded data.

Comment 3: Reservoir catchment details have not been presented appropriately. A full section should be added in the manuscript explaining the reservoir details and their primary usage (i.e., flood regulation, hydro-power generation, irrigation purpose etc.). Besides, Figure S1 should be included in main document with more clarity.

Comment 4: It would be better to included more details of Kerala flooding and some of the satellite images (RADARSAT, MODIS etc.) to show flood extent, inundation depth, and time along with the incurred financial losses with lost human lives and livestock to show the severity of Kerala floods. In this way a wider audience from other part of the world which is not aware of this calamity would be able to relate it easily. As of now poor description makes it very difficult for people outside of India to comprehend the Kerala flood.

Minor comments:

[Figure]

P2 - L-18: Does IMD already providing quarter degree rainfall data for year 2018? As per the norms the gridded data for year 2018 would be released next year. Please confirm this.

P2 - L 19-20: IMD data has been prepared using 6000 rain gage data gives incomplete picture and a misleading statement as all these station never used to prepare the gridded data simultaneously. Moreover, "substantial number of stations are located in Kerala" does not help much. Authors should rather provide the year wise number of functional stations that have been used in the preparation of gridded data for better understanding including the spatial distribution.

P2 – L3 How many year of data was used to estimate the distribution parameters.

P3 – L18 How does authors justify the usage of 117 years of data to obtain the long term average of rainfall for Kerala state. Is the long term rainfall is the true representation of average rainfall that Kerala receives owing to the changing climate. Is it the right approach to go for longest available time series for obtaining mean characteristics?

P7 – L1 Authors should present the number from rain gauge stations lying in the reservoir catchment instead of using grid based values. The station data can be easily obtained from IMD and would be much accurate as gridded data is often have smoothening effect due to interpolation.

---

## Short Comment (SC3) · 29 Oct 2018

Owing to the frequent flooding being reported at various part of the world it is of prime importance to study such events carefully to minimize the brunt of such calamities in future event. Though I do feel that topic is very interesting for the scientific communities working in the field of hydrology and climatology, I am also of the opinion that a sufficient in-depth diagnostic of this interesting case study is not up to the mark hence, does not provide the important facts and details comprehensively.

The authors mentioned the large-scale flooding in Kerala, India, during August 2018, which affected millions of people and caused 400 or more deaths. This shows heavy loss to human lives as well as the property. This Kerala flood event could be the

worst flood of India (not only Kerala) in the century and may be well known globally by the scientific leaders working in the related research area. Here my concern is that, the manuscript does not provide any map/figure which shows a clear picture of disaster caused by climate change event or specific atmospheric phenomenon that triggered the event. The authors have provided only few statements mentioning about Kerala flood due to extreme rainfall event followed by the past flood events occurred in India. In my opinion, the research papers or technical notes should be scientifically sound than the articles available from electronic media. The manuscript should follow a holistic approach for better understanding to the scientific readers. Here, the authors should have provided the map showing the extent of flood or flood inundation for a better understanding of the readers those were not aware about this event. Moreover, this flood event as well as the extreme rainfalls occurred across all the catchments or Kerala state should have been compared with similar other events that took place in the past at other part of the India sub-continent in general and India in particular.

The representation of the study area is also very poor. The authors are explaining about the reservoirs but did not show the catchment of any of these reservoirs in the main document and rather put in the supplementary material. Besides, I could not find any major details of any of these reservoirs (such as how big they are etc.). I could not understand the purpose of showing Fig. 1c, where cumulative rainfall over whole Kerala state is presented. Showing cumulative rainfall over the catchment area of each reservoir would be a better approach that supposedly resulted in the heavy flooding in the selected catchments.

The manuscript is also lacking the physical reasoning for the heavy flooding; and only statistical analysis has been presented in the manuscript towards extreme rainfall and reservoirs storage. Also it is very difficult to understand from the manuscript about the purpose of these dams/ reservoirs. Is the major purpose of these dams to control the flood or full filling the irrigation requirement of the state? It is worth mentioning about the reservoir operation. If most of these are flood controlling then, why were they failed

in controlling the floods? The authors should also explain these small but important facts to support the statistics resulted from the hydro-climatic dataset.

Due to lack of the information about this recent disaster event in the manuscript I tried to find more details on google for my own understanding. One of the report published in 'Down to Earth' discusses lack of emergency plan in about 61 dams however, the authors have selected only seven reservoirs. It will be very interesting for the readers, if the authors explain about the basis for selecting these seven reservoirs and why other dams are not so important and should not be the part of such analysis. It would also be interesting, if authors can show how the floods could have been prevented if these seven reservoir had been regulated properly leaving other 54 reservoir/dams.

The manuscript uses the IMD gridded rainfall product which is developed thorough Inverse Distance Weighting interpolation method using station based rainfall observations. Why the authors have used this gridded rainfall product which has interpolated information at 0.25°. In my opinion, the gridded rainfall may lose extreme information after interpolation. The use of station data rather than gridded products could strengthen the statistic obtained in the reported study. As the authors have not provided any details regarding catchment size of the selected reservoirs, it is difficult to figure out the geographical area contributing to the selected reservoirs; if the catchment area of these reservoirs is small authors should have used station data under each catchment instead of interpolated data at 0.25°. Also representation of the rain gauges locations in each catchment is also necessary. It is my humble suggestion to the authors to provide detailed information in figures to improve the clarity of the paper which will help the young scientists in better understanding.

Furthermore, the present statistics are not sufficient enough to define this large scale disaster or we can say that there may be number of other reasons which were responsible for this heavy flood. The poor description of the actual process may mislead the future studies in-line this case study. I suggest the authors to use appropriate rainfall dataset, conceptualize the whole process and finally go for the concrete conclusion.

The authors can take the help of some well-established hydrological models for physical understating of the whole hydrological process.

Overall, the topic is very interesting and eye caching but manuscript does not lead to any scientific advancement. "Hydrology and Earth System Science" is reputed in the scientific community for publishing technical notes/report with new developments, novel approaches, and techniques relevant for scientific investigations, however this manuscript fails to present any scientific advances in terms of theoretical methods or techniques. Therefore, I request authors to kindly take notes of the comments and revise the manuscript accordingly. I strongly believe that this study has the potential of becoming an important piece of work once it is revised properly considering all the facts and details.

---

## Author Comment (AC1) · 4 Dec 2018

This paper and its findings are very relevant in the context of extreme climatic events recurring at an alarming rate all over the world. A critical analysis of Kerala flood of 2018 presented here must be of great interest to the researchers, water resources managers as well as the development practitioners. Authors have done a good job and the presentation of its results are appealing to the scientific community.

Thanks for your positive comments on our manuscript.

The estimation of return period of extreme rainfall using GEV distribution and its analysis are well presented. In that aspect, this paper is an outstanding work. However, there are certain important aspects missed out in the analysis of reservoirs' storage and that makes the conclusions drawn a bit unscientific. I would request the authors to consider the following aspects and rework on it to make the paper more applicable to a wider section of readers among researchers and practitioners.

(a) Unlike the impact of extreme rainfall analysed and presented, the impact of reservoir storage in worsening the flood situation appears to be only a general statement and it is not supported with a clear and logical analysis. (b) Assessing the impact of reservoir storage by just looking at the percentage of FRL storage is a premature analysis.

This has to be analysed in a more logical manner by checking the rise in water level caused at a downstream point due to reservoir releases. To be very specific, you need to do this analysis for at least 5 major reservoirs you are considering. Idukki, Idamalayar and Periyar in the Periyar basin, Malampuzha in the Bharathapuzha basin and Parambikulam in the Chalakkudipuzha basin. CWC gauging stations are available on the downstream side of these reservoirs in Periyar, Chalakkudipuzha and Bharatha-puzha. So the flow data at these points are already there in the public domain.

Thanks for your insightful suggestion. In the revised manuscript, we provide more analysis pertaining to the role of reservoirs in the flooding by analyzing the downstream streamflow data as well as the reservoir release data. CWC provides only reservoir storage but not the release information. The reservoir release data is not available in the public domain and almost impossible to get the data due to regulation. Moreover, we try to include the discussion on the reservoir rule curve to support our analysis.

Simulate the depth of flow at these points without the reservoir releases and with reser-voir releases and see whether the impact is significant (reservoir release data is also in the public domain, or you can get it from the respective dam managers). If you are getting the impact significant, simulate again the reservoir storage to safer position and see the volume that should have been left unfilled in the reservoir to make the impact of its releases insignificant. This finding is also important. I would expect such an analysis before jumping into a conclusion that reservoirs storage also contributed to the flood.

Since the reservoir release data is not available, it is almost impossible to generate the effect of reservoir release on the downstream flood. However, we discuss this in more details based on the information we have. We include the SWAT model simulations for a few major catchments to provide more information on the streamflow conditions upstream of the reservoirs and downstream in the context of reservoir rule curves.

Another important point to be considered in this analysis is that the main objective of major reservoirs you have taken is water conservation and not flood control. When you optimize the storage based on the major objective of water conservation, what is the limitation on unfilled volume should also be clearly investigated before making a conclusion.

On this aspect, we provide a discussion in the revised manuscript. However, since our technical note is not on the simulation of flood conditions and the role of reservoirs on that, we limit our analysis on the extreme value theory on comment/discuss on the role of reservoirs based on the information available. We wish to provide more details, however, data availability is a major problem and obtaining the data for the detailed analysis is extremely challenging. Notwithstanding these limitations, we provide a first assessment on the extreme rainfall and simulated flow conditions during the Kerala flood of 2018.

Hope the authors would consider these points and revise the MS with these results to make it more convincing. This paper would be of interest to the broader public if the authors take up the additional analysis I have suggested and revise the MS accordingly.

Thanks. We have made a serious effort to revise the manuscript given that the data limitations.

---

## Author Comment (AC2) · 4 Dec 2018

Note to the editor and authors: As part of an introductory course to the Master programme Earth Environment at Wageningen University, students get the assignment to review a scientific paper. Since several years, students have been reviewing papers that are in open online discussion for HESS or BGS, and they have been asked to submit their reports to the discussion in order to help the review process. While these reports are written in the form of official (invited) reviews, they were not requested for by the editor, and we leave it up to the editor and authors to use these reports to their advantage. While several students were often asked to review the same paper, this was not done with the aim to provide the authors with much extra work. We hope that these reports will positively contribute to the scientific discussion and to the quality of papers published in HESS. This report/review was supervised by dr. Ryan Teuling (teacher within the ITEE course at Wageningen University and also associated editor with HESS).

We really appreciate this amazing effort and learning experience. We consider all the comments positively and will address them in the revised version. Thanks to all the students and Dr. Teuling for this excellent review.

This study is aimed at finding the causes of the Kerala flood in August 2018. A gridded rainfall map of India was constructed for the analyses and data of major reservoirs in Kerala were gathered. Based on the available data reservoir characteristics, mass curves and depth-duration-frequency curves were constructed. Results showed unprecedented amounts of rainfall in catchments upstream of major reservoirs with return periods much larger than 500 years. It was also shown that most major reservoirs were almost at their full reservoir level at the start of the series of extreme rainfall events. It is concluded that both the high reservoir storage and extreme amounts of rainfall played a significant role in the large-scale flooding in Kerala. It is advised to improve forecasts of extreme rain at longer lead times to better manage reservoir operations. Research in this field has a high societal relevance since a better understanding of the causes of major floods, like in Kerala, can save future lives. A similar study, Sayama et al. (2012), also stresses the importance of this kind of analyses to better understand and prevent future floods. The study is also relevant since similar events are predicted to occur more often in India under the effect of global warming. However, in my opinion the authors did not conduct a sufficiently in depth study of this interesting case. Results are so extreme that validity of the data has to be questioned. Moreover, a proper discussion about these extreme outcomes is missing. Furthermore, some figures are missing or current figures need to be altered to improve the clarity of the paper. In spite of its high relevance, I would recommend that major revisions are necessary before the work can be published in HESS.

Thanks for the insightful suggestions. As suggested by the two reviewers, we have added more discussion and analysis in the revised version. Moreover, we have carefully checked the precipitation and other datasets used in the manuscript during the revision.

**Major comments**

(1) The paper concludes that rainfall in the catchments upstream of major reservoirs was unprecedented. It is stated that Idukki, Kakki and Periyar reservoir experienced 279, 700, and 420% surplus rainfall, respectively from their long-term mean between May 1 and August 21 in 2018. These values are so extreme that I highly doubt the correctness of the data. Moreover, the discussion section does not mention anything about these seemingly unrealistic results.

Thanks. We have evaluated the rainfall record again during the revision. The rainfall data is based on the gage stations of India Meteorological Department (IMD). We have collected observed station data to compare with the gridded data.

These conclusions are based on figures 3 and 4. Idukki, Kakki and Periyar reservoirs clearly stand out from the rest. In figure 3 the upstream catchments of Idukki, Kakki and Periyar reservoirs show cumulative rainfall amounts after three months that are already 2-3 times higher than the long-term average amount of rainfall for an entire year in these catchments. Furthermore, the cumulative amount of rainfall in these three catchments is far away from the grey area indicating the area within one standard deviation from the long-term mean. In figure 4 the upstream catchments of Idukki, Kakki and Periyar reservoirs show rainfall amounts during August 2018 for a duration of 15 days that are 2-3 times as high as the amount that would occur once every 500 years. Rainfall amounts depicted in figures 3 and 4 are so unprecedented that it is highly unlikely that these amounts actually occurred.

As stated above, we have carefully checked all the dataset and analysis in the revised manuscript to avoid any error.

Major alterations are necessary to sufficiently improve the discussion section and the results shown in figures 3 and 4. I would recommend to reconsider the data used to construct the figures. From the methods it is not clear to me whether a quality control has been conducted on the raw data. If not the case, I would strongly recommend to do one as this might eliminate outliers that cause the seemingly impossible results in figures 3 and 4. The quality control used in Yatagai et al. (2012) and Haylock et al. (2008) might be appropriate methods. Also, the discussion section should be expanded with a discussion about these extreme results.

Thanks. The two reviewers also suggested to improve the discussion. In the revised manuscript, we provide in-depth discussion on the simulated flow conditions, reservoir operations, the role of large-scale climate, and the role of extreme precipitation forecast. We used the standard gridded precipitation product from IMD, which has been evaluated against the other precipitation products (APHRODITE, Yatagai et al. (2012)). We have mentioned this in the revised manuscript.

(2) Major conclusions are solely based on data obtained from rain gauges in Kerala.

No information is given about the location of these gauges. Moreover, I am missing information about the size of the catchments upstream of the major reservoirs and the location of rain gauges in these catchments and their surrounding areas. Not including this information gives an incomplete picture of where the data comes from and what the data actually describes. Validation of the data could help in understanding how questionable results, like addressed in my previous point, can occur.

We provide more information on the number of rainfall gauge stations as well as the catchment area in the revised manuscript. We compare the gridded precipitation with the Global Precipitation Mission (GPM) as well as monthly station-based rainfall observations.

To improve the clarity of the paper I would first of all advise to include a map of all rain gauges in India, like is given in Pai et al. (2014a). In this map Kerala should be clearly delineated. By including this map readers of the paper will be able to see the density and

distribution of rain gauges in India and Kerala. Furthermore, I think it is essential that separate maps of all reservoirs and their upstream catchments should be included. In this map the location of rain gauges in the catchment and its surroundings should be clearly indicated. Including these maps would contribute in understanding how the rainfall data of the catchments is obtained.

Thanks. We have used the rainfall station map in the supplemental section of the revised manuscript.

(3) In the paper variability is estimated using one standard deviation. One standard deviation indicates the variability in which 68% of the values will fall. This is meaningless since values outside this variability cannot be regarded as extreme.

Indicating variability with two standard deviations would be an improvement. 95% of all values will fall within a variability of two standard deviations, values outside this variability can now be regarded as extreme. This especially holds for figures 1a, 2 and 3. For example, extending the variability indicated by the grey area in figure 2 to two standard deviations would cause most values of 2018 to fall within this variability. Values that are outside this area can now be regarded as extreme, this would add to the strength and clarity of the figure.

Thanks. We have revised the plots using both one and two-standard deviations.

**Specific comments**

(1) Be consistent with writing down dates throughout the document.
Done

(2) Use "that" before "occurred" across the document like in p2 line 1, 9 and 32.
    Done

(3) The statement made in p3 line 9 seems very extreme and I think these amounts of rainfall are highly unlikely to have occurred in large parts of Kerala. Also, when looking at the DDF curves this will result in gigantic return periods. Based on what is this statement based? I would remove this statement.
We have modified the statement, thanks.

(4) p2, line 1: missing "that" before "occurred"
    Done

(5) p2, line 6: "Dottori et al. (2018)" instead of "Dottori et al., (2018)"
    Done

(6) p2, line 6: "an" before "uneven"
    Done

(7) p2, line 9: missing "that" before "occurred"
    Done

(8) p2, line 19: missing "the" before "IMD"
    Done

(9) p2, line 19: Cite Pai et al. (2014a) here since the dataset has been developed based on the methods described in this paper.
Done

(10) P2, line 22: The wrong citations are used here. Mishra et al. (2014) only cites a paper which makes this statement, Shah and Mishra (2015) does not make this statement at all. The only correct reference here should be Pai et al. (2014b).
    Done

(11) P2, line 24 and 25: It is stated that "Gridded daily rainfall from IMD has been widely used in hydro-climatic studies" with five references to support this claim. However, all references are papers published by one of the authors. Add references which are not of one of the authors or do not state that it is "widely used".
Thanks. Done

(12) p2, line 26 and 27: "curves" instead of "curve"
Done

(13) p2, line 31: "a" before "once"
Done

(14) p2, line 32: "The" before "return"
Done

(15) p3, line 6: "periods" instead of "period"
Done

(16) p3, line 6: "the" before "GEV"
Done

(17) p3, line 6: "a" before "Chi-square"
Done

(18) p3, line 6: replace "test, and we" by "test. We"
Done

(19) p3, line 7: remove the third "the"
Done

(20) p3, line 12: "IWRIS" instead of "WRIS"
Done

(21) p4, line 2 and 3: I would not use percentages, they are quite meaningless here in the way it is written down. Better use exact numbers or reformulate.
Done

(22) P4, line 7: "the" before "mean"
Done

(23) P4, line 8: remove last ","
Done

(24) The statement made in p4 line 9 seems very extreme and I think these amounts of rainfall are highly unlikely to have occurred in large parts of Kerala. Also, when looking at the DDF curves this will result in gigantic return periods. Based on what is this statement based? I would remove this statement.
Done

(25) p4, line 16: the reservoir dataset from 2007-2017 is referred to as a long-term mean, I would not call 11 years of data a long-term dataset.
Done

(26) p4, line 14: Remove double spacing before "We"
Done

(27) p5, line 3: "respectively" before "279"
Done

(28) p5, line 23: "2018" before "in"
Done

(29) p5, line 28: "the" before "future"
Done

(30) p5, line 30: "1-5 day" instead of "1-5day"
Done

(31) p5, line 31: remove "the"
  Done

(32) p6, line 8: "and" before second "land"
  Done

(33) p6, line 12: "than" instead of "that"
  Done

(34) p6, line 13: "periods" instead of "period"
  Done

(35) p6, line 19: "of" before "extreme"
  Done

(36) p6, line 26: "periods" instead of "period"
  Done

(37) p6, line 28: remove "the"
  Done

(38) p6, line 28: "periods" instead of "period"
  Done

(39) p6, line 29: "were at" instead of "had"
  Done

(40) p7, line 7: "time" before "can"
  Done

(41) p7, line 7: "improving" before "reservoir"
  Done

(42) Figure 1a: I would indicate the start of events on the 7th of August in the same way as in figure 3 with a thin line. Indication now looks like a weird dip in the data.
Done

(43) Figure 1 caption: state that the delineated part in India is Kerala.
Done

**References**

Haylock, M. R., Hofstra, N., Tank, A. K., Klok, E. J., Jones, P. D., New, M. (2008). A Eu-ropean daily high-resolution gridded data set of surface temperature and precipitation for 1950–2006. Journal of Geophysical Research: Atmospheres, 113(D20).

Pai, D. S., Sridhar, L., Rajeevan, M., Sreejith, O. P., Satbhai, N. S., Mukhopadhyay, B. (2014a). Development of a new high spatial resolution (0.25× 0.25) long period (1901–2010) daily gridded rainfall data set over India and its comparison with existing data sets over the region. Mausam, 65(1), 1-18.

Pai, D. S., Sridhar, L., Badwaik, M. R., Rajeevan, M. (2014b). Analysis of the daily rainfall events over India using a new long period (1901–2010) high resolution (0.25× 0.25) gridded rainfall data set. Climate dynamics, 45(3-4), 755-776.

Sayama, T., Ozawa, G., Kawakami, T., Nabesaka, S., Fukami, K. (2012). Rainfall–runoff–inundation analysis of the 2010 Pakistan flood in the Kabul River basin. Hydro-logical Sciences Journal, 57(2), 298-312.

Yatagai, A., Kamiguchi, K., Arakawa, O., Hamada, A., Yasutomi, N., Kitoh, A. (2012). APHRODITE: Constructing a long-term daily gridded precipitation dataset for Asia based

on a dense network of rain gauges. Bulletin of the American Meteorological Society, 93(9), 1401-1415.

Thanks. We have cited the relevant references.

---

## Author Comment (AC3) · 4 Dec 2018

The manuscript 'The Kerala flood of 2018: combined impact of extreme rainfall and reservoir storage' by Mishra and others includes very interesting argument on recent flood in the Kerala. The topic is very interesting and of great interest to the scientific communities working in the field of climatology and hydrology as well. However, certain portion of the manuscript needs substantial reworking before it can be referred in HEES. The research purpose, significance and objectives are clearly stated but not well organized. More detailed and accurate descriptions are required. I suggest some parts of abstract would fit better into the introduction. Moreover, authors should begin the abstract directly with results/findings, and/or state that the area received several anomalous rainfall storms throughout the past and recently experienced 53% above normal rainfall (Monsoon season 2018).

Thanks for your positive remarks and suggestions. We have carefully revised the manuscript and incorporated the comments.

The rainfall in the Kerala is predominately controlled by the South-west and North-east monsoons. The area has witnessed heavy losses to life and property by floods in almost all rivers of Kerala due to several rainstorms in the past as well. Despite the fact such information is missing in the text and fails to convey a clear and convincing introduction. I suggest that keep one paragraph in introduction about the history of floods in Kerala and discuss the conditions about reservoir operations during such rainfall events. The contents of this paper are valuable, but the authors should pay close attention to the data sets and results. Authors have used rigorous statistical methods to compare peak monsoon rainfall patterns during two time periods. The team looked specifically at rainfall during the month of August, which is the peak of the monsoon. Further, explain the percent (%) of rainfall over Kerala during June, July and 1st to 19 th of August, above selected normal. Catchment area of each sub-basin is believed to be calculated and should be included in the text, and therefore comparison of rainfall depths observed in different sub-basins and rest of the Kerala during event will be computed.

Thanks. We have included a paragraph on the historical floods in Kerala and their deriving factors such as the role of reservoirs. We have provided a table with catchment areas and rainfall depth in the revised manuscript.

Rainfall received during the summer monsoon season contributes about 70–90% of annual rainfall over India. The intensity and magnitude of these floods are the manifestation of year to year variability of monsoon over India. Also, it has been recognized that such variability of ISM has a good teleconnective relation with El Nino Southern Oscillation (ENSO), North Atlantic Oscillation and climate extreme indices available. Therefore, team should look at the effects of teleconnection patterns (TPs) on the extremes of precipitation over Kerala. Whether, the patterns of extreme wet and dry spells during the monsoon season have changed in recent decades (1901-2018). An understanding of the teleconnection patterns associated with these events could benefit many people and policy makers in the state. It is essential to re-evaluate the operating criteria, guidelines that govern the storage and release functions of a reservoir in such extreme conditions. You would need to have much more discussion highlighting how

your results are relevant for climate change adaptation and disaster risk reduction in the region. This has not really been demonstrated in discussion or conclusions.

Thanks. We have added a new section on the sea surface conditions in the Indian and Pacific Oceans to highlight the potential role of large-scale climate drivers. Moreover, we have included discussion on the reservoir rule curves in the revised manuscript. Since the reservoir release data is not in the public domain and almost impossible to get due to regulations, we limit our discussion on the available datasets and develop insights from them.

Can you expand on this? "Reservoir operations need to be improved using a skillful forecast of extreme rainfall at the longer lead time (4-7 days)". Your result shows that, if the reservoir had been below FRL, the flooding conditions would have not changed much due to the severe storm continued for 3-4 days. It would have been necessary to release from the reservoirs after 1st day of the extreme rainfall. Therefore, improved forecast of extreme rainfall from onset of the monsoon along with reservoir conditions (Inflow and outflow) must be reviewed and design accordingly. The probable maximum flood (PMF) is frequently revised as the required inflow design flood (IDF) until all the necessary safety conditions will be satisfied.

Thanks. We have provided a detailed discussion on the role of improved extreme precipitation forecast on reservoir operations. We have cited the studies that used such information in other countries, which highlight the need of an improved forecast.

Results and conclusions are almost similar. Findings you infer from your data should be worked out more thoroughly in discussion section. Try to be more stringent when presenting your data and avoid repletion of similar sentences. Pleas find more details in the attached file.

Thanks. As mentioned above, we have provided in-depth discussion and included a new section on the large-scale climatic factors in the revised manuscript.

The manuscript needs to substantially improve in English. I think manuscript need thorough revision for it to get to the standard that it deserves. I strongly encourage authors to read these comments without any pre-conceived notion because I think this should be published once revised appropriately.

Thanks. We have checked the revised version for English and Grammar.

---

## Author Comment (AC4) · 4 Dec 2018

In the submitted manuscript authors made an attempt to analyze the rainfall and reservoir level in the monsoon months of prior August 2019. Apparently, many of the findings that authors have claimed are already available in public domain perhaps not in the same way the team has presented. I have included only some of the links below for reference.

1. https://www.firstpost.com/tech/news-analysis/what-caused-the-kerala-floods-4993041.html 2. https://indianexpress.com/article/research/year-1099-keralas-great-flood-of-1924-too-affected-same-areas-5317677/
3. https://www.bloombergquint.com/kerala-floods/kerala-flood-of-2018-less-intense-than-deluge-of-1924-so-why-was-damage-as-great 4. https://timesofindia.indiatimes.com/india/in-just-20-days-kerala-gets-highest-aug-rains-in-87-yrs/articleshow/65480279.cms 5. https://www.indiatoday.in/india/story/why-kerala-fears-repeat-of-1924-havoc-in-2018-rainfall-1315884-2018-08-16 6. https://scroll.in/article/890593/monsoon-trends-for-many-in-kerala-this-years-rains-recalls-the-great-flood-of-99

Apparently, some of the articles appeared in print and electronic media made a more comprehensive and holistic overview of Kerala flooding, I assume, keeping the data analysis in the background. Please excuse me if I am being blunt or it reads harsh, as a reader of Hydrology and Earth System Science I would always seek a bit more scientific content with cutting edge hydrological analysis from a HESS article. The submitted manuscript is undeniably a good piece of work when I see it as a term project where grad students are asked to perform a quick analysis of rainfall pattern and reservoir levels to reason with the flooding, however, for a scientific paper it requires some more meat.

At its current form the submitted manuscript can be accepted in conference proceedings but for being considered as a potential HESS journal article it requires major re-vision with some additional analysis. To avoid overlap in comments made in 'Referee Comment 2' and 'Short Comment 1', I am not including the common queries here and only including those parts which I find missing or less emphasized in their observations.

Since, I am not the assigned referee of the paper authors are free to discard my comments, however, I would like to state few of my concerns that I would like to addressed by authors to make the manuscript more elaborate, scientific, and a good hydrology paper rather than being a mere data analysis.

Thanks for your comments. Both the reviewers and short-comment #1 suggest the importance of the work while they recommend more discussion on the certain aspects. The work available in the media does not provide in-depth discussion and scientific analysis.

**Major comments:**

Comment 1: The flood extent and inundation through hydrodynamic modeling approaches in the downstream of various reservoirs taken in the study will provide more insight into the problem. While doing so if authors can present a comparative analysis of flood extent between two scenarios as follows. Scenario 1: Flood extent with actual reservoir levels. Scenario 2: Flood extent if the reservoirs would have been regulated properly before the

heavy rainfall hit the catchment i.e., the best recommended practice. Though it will take some good amount of effort, it will add enormous value to manuscript.

We understand the importance of your suggestion. However, this is clearly beyond the scope of our manuscript. We aim to present the in-depth analysis on the extreme rainfall that caused flood in Kerala. However, entire hydraulic analysis and simulation of flood inundation need significant more work and data requirements. We do not have precise cross section information neither do we have reservoir release data. We only have rainfall and reservoir storage data to analyze the Kerala flood of 2018. Notwithstanding, the importance of the suggestions, we are unable to include all the aspects of the Kerala flood in this manuscript.

Comment 2: Did authors selected the specific IMD grid in which reservoir falls to analyze the rainfall or utilize the catchment average rainfall calculated for IMD gridded rainfall. Gridded data is prepared with the varying network of rain gages hence there could be inhomogeneity in the time series. Data constraint can be accepted while doing such analysis over India scale however, for small catchments only station data should be preferred. Since, this is a very localized study, it would be more appropriate if authors use station data instead of gridded rainfall data. Authors should include the rain gauge network from the reservoir catchment and utilize the rain gage data instead of gridded data.

We analyze two datasets: 1) standard 0.25 deg gridded rainfall data from IMD, and 2) precipitation data from global precipitation mission (GPM). These both the datasets are consistent. Getting daily rainfall data from individual stations is challenging and may not be error free. We will make efforts to obtain such data, if possible, and include that in the revised manuscript.

Comment 3: Reservoir catchment details have not been presented appropriately. A full section should be added in the manuscript explaining the reservoir details and their primary usage (i.e., flood regulation, hydro-power generation, irrigation purpose etc.). Besides, Figure S1 should be included in main document with more clarity.

Thanks. We have provided more details about the reservoir catchments in the revised manuscript.

Comment 4: It would be better to included more details of Kerala flooding and some of the satellite images (RADARSAT, MODIS etc.) to show flood extent, inundation depth, and time along with the incurred financial losses with lost human lives and livestock to show the severity of Kerala floods. In this way a wider audience from other part of the world which is not aware of this calamity would be able to relate it easily. As of now poor description makes it very difficult for people outside of India to comprehend the Kerala flood.

Thanks. We have included satellite imagery in the revised version to show the extent of flooding.

**Minor comments**
P2 - L-18: Does IMD already providing quarter degree rainfall data for year 2018? As per the norms the gridded data for year 2018 would be released next year. Please confirm this.

Yes, we have obtained the data from IMD.

P2 - L 19-20: IMD data has been prepared using 6000 rain gage data gives incomplete picture and a misleading statement as all these station never used to prepare the gridded

data simultaneously. Moreover, "substantial number of stations are located in Kerala" does not help much. Authors should rather provide the year wise number of functional stations that have been used in the preparation of gridded data for better understanding including the spatial distribution.

Thanks. In the revised manuscript, we have provided the stations that were used to obtain the gridded rainfall data from IMD.

P2 – L3 How many year of data was used to estimate the distribution parameters.

We used entire data (1901-2017) to estimate the parameters of the distribution.

P3 – L18 How does authors justify the usage of 117 years of data to obtain the long term average of rainfall for Kerala state. Is the long term rainfall is the true representation of average rainfall that Kerala receives owing to the changing climate. Is it the right approach to go for longest available time series for obtaining mean characteristics?

This is probably the best dataset any one can use. I am not aware of any other long-term data for the state of Kerala.

P7 – L1 Authors should present the number from rain gauge stations lying in the reservoir catchment instead of using grid based values. The station data can be easily obtained from IMD and would be much accurate as gridded data is often have smoothening effect due to interpolation.

Thanks, we have included this in the revised manuscript.

---

## Author Comment (AC5) · 4 Dec 2018

Owing to the frequent flooding being reported at various part of the world it is of prime importance to study such events carefully to minimize the brunt of such calamities in future event. Though I do feel that topic is very interesting for the scientific communities working in the field of hydrology and climatology, I am also of the opinion that a sufficient in-depth diagnostic of this interesting case study is not up to the mark hence, does not provide the important facts and details comprehensively.

Thanks. As suggested by the reviewers, we have included a detailed discussion on the following aspects in the revised manuscript: 1) nature of extreme rainfall, 2) the role of simulated flow upstream of the reservoirs, 3) the role of reservoirs and their rule curves, 4) the influence of SST condition prior to the extreme rainfall. We feel that after including these aspects in the revised manuscript, manuscript will improve significantly to provide a comprehensive assessment on the Kerala Flood of 2018.

The authors mentioned the large-scale flooding in Kerala, India, during August 2018, which affected millions of people and caused 400 or more deaths. This shows heavy loss to human lives as well as the property. This Kerala flood event could be the worst flood of India (not only Kerala) in the century and may be well known globally by the scientific leaders working in the related research area. Here my concern is that, the manuscript does not provide any map/figure which shows a clear picture of disaster caused by climate change event or specific atmospheric phenomenon that triggered the event. The authors have provided only few statements mentioning about Kerala flood due to extreme rainfall event followed by the past flood events occurred in India. In my opinion, the research papers or technical notes should be scientifically sound than the articles available from electronic media. The manuscript should follow a holistic approach for better understanding to the scientific readers. Here, the authors should have provided the map showing the extent of flood or flood inundation for a better understanding of the readers those were not aware about this event. Moreover, this flood event as well as the extreme rainfalls occurred across all the catchments or Kerala state should have been compared with similar other events that took place in the past at other part of the India sub-continent in general and India in particular.

Thanks for your valuable suggestions. As mentioned above, we provide a discussion on the potential role of large-scale climate variability in the extreme precipitation using SST condition in the Indian and Pacific Oceans. However, we feel that if the event was caused by climate change or not, remains beyond the scope of this manuscript. We have no doubt that this is an important aspect and need to be studied. Our aim was not to compare this event with the flooding in the other parts of the region.

The representation of the study area is also very poor. The authors are explaining about the reservoirs but did not show the catchment of any of these reservoirs in the main document and rather put in the supplementary material. Besides, I could not find any major details of any of these reservoirs (such as how big they are etc.). I could not understand the purpose of showing Fig. 1c, where cumulative rainfall over whole Kerala state is presented. Showing cumulative rainfall over the catchment area of each reservoir would be a better approach that supposedly resulted in the heavy flooding in the selected catchments.

Thanks. We have included more details on the study area with the figure in the main manuscript. We provide a separate table on the details of the reservoirs.

The manuscript is also lacking the physical reasoning for the heavy flooding; and only statistical analysis has been presented in the manuscript towards extreme rainfall and reservoirs storage. Also it is very difficult to understand from the manuscript about the purpose of these dams/ reservoirs. Is the major purpose of these dams to control the flood or full filling the irrigation requirement of the state? It is worth mentioning about the reservoir operation. If most of these are flood controlling then, why were they failed

in controlling the floods? The authors should also explain these small but important facts to support the statistics resulted from the hydro-climatic dataset.

Thanks. As suggested by the other reviewers, we have included the analysis on the simulated flow and reservoirs operations (based on rule-curves) in the revised manuscript.

Due to lack of the information about this recent disaster event in the manuscript I tried to find more details on google for my own understanding. One of the report published in 'Down to Earth' discusses lack of emergency plan in about 61 dams however, the authors have selected only seven reservoirs. It will be very interesting for the readers, if the authors explain about the basis for selecting these seven reservoirs and why other dams are not so important and should not be the part of such analysis. It would also be interesting, if authors can show how the floods could have been prevented if these seven reservoir had been regulated properly leaving other 54 reservoir/dams.

We do not have observed data for all the reservoirs. Central Water Commission (CWC) provides observations for only seven major reservoirs located in Kerala. Therefore, we used only major reservoirs in our analysis. We have mentioned this in the revised manuscript.

The manuscript uses the IMD gridded rainfall product which is developed thorough Inverse Distance Weighting interpolation method using station based rainfall observations. Why the authors have used this gridded rainfall product which has interpolated information at $0.25^{\circ}$. In my opinion, the gridded rainfall may lose extreme information after interpolation. The use of station data rather than gridded products could strengthen the statistic obtained in the reported study. As the authors have not provided any details regarding catchment size of the selected reservoirs, it is difficult to figure out the geographical area contributing to the selected reservoirs; if the catchment area of these reservoirs is small authors should have used station data under each catchment instead of interpolated data at $0.25^{\circ}$. Also representation of the rain gauges locations in each catchment is also necessary. It is my humble suggestion to the authors to provide detailed information in figures to improve the clarity of the paper which will help the young scientists in better understanding.

We have added more information related to rain-gauge stations in the revised version. Moreover, we are trying to obtain observed rainfall data based on stations to compare with the gridded data. Long-term station based observations that are required for extreme value analysis are largely unavailable. Therefore, we used gridded rainfall data from IMD.

Furthermore, the present statistics are not sufficient enough to define this large scale disaster or we can say that there may be number of other reasons which were responsible for this heavy flood. The poor description of the actual process may mislead the future studies in-line this case study. I suggest the authors to use appropriate rainfall dataset, conceptualize the whole process and finally go for the concrete conclusion.

Thanks. We have revised the manuscript and provided more discussion. However, we can not evaluate all the aspects of flooding due to data limitations, which has been mentioned in the conclusion section.

The authors can take the help of some well-established hydrological models for physical understating of the whole hydrological process.

Thanks. We have used SWAT model to simulate the daily flow upstream of major reservoirs.

Overall, the topic is very interesting and eye caching but manuscript does not lead to any scientific advancement. "Hydrology and Earth System Science" is reputed in the scientific community for publishing technical notes/report with new developments, novel

approaches, and techniques relevant for scientific investigations, however this manuscript fails to present any scientific advances in terms of theoretical methods or techniques. Therefore, I request authors to kindly take notes of the comments and revise the manuscript accordingly. I strongly believe that this study has the potential of becoming an important piece of work once it is revised properly considering all the facts and details.

Thanks for your constructive comments.